# A Metzincin and TIMP-Like Protein Pair of a Phage Origin Sensitize *Listeria monocytogenes* to Phage Lysins and Other Cell Wall Targeting Agents

**DOI:** 10.3390/microorganisms9061323

**Published:** 2021-06-18

**Authors:** Etai Boichis, Nadejda Sigal, Ilya Borovok, Anat A. Herskovits

**Affiliations:** The Shmunis School of Biomedicine and Cancer Research, Faculty of Life Sciences, Tel Aviv University, Ramat Aviv, Tel Aviv 69978, Israel; eboichis@gmail.com (E.B.); sigaln@tauex.tau.ac.il (N.S.); ilyabo@tauex.tau.ac.il (I.B.)

**Keywords:** *Listeria monocytogenes*, phage, metzincin, TIMP

## Abstract

Infection of mammalian cells by *Listeria monocytogenes* (*Lm*) was shown to be facilitated by its phage elements. In a search for additional phage remnants that play a role in *Lm*’s lifecycle, we identified a conserved locus containing two XRE regulators and a pair of genes encoding a secreted metzincin protease and a lipoprotein structurally similar to a TIMP-family metzincin inhibitor. We found that the XRE regulators act as a classic CI/Cro regulatory switch that regulates the expression of the metzincin and TIMP-like genes under intracellular growth conditions. We established that when these genes are expressed, their products alter *Lm* morphology and increase its sensitivity to phage mediated lysis, thereby enhancing virion release. Expression of these proteins also sensitized the bacteria to cell wall targeting compounds, implying that they modulate the cell wall structure. Our data indicate that these effects are mediated by the cleavage of the TIMP-like protein by the metzincin, and its subsequent release to the extracellular milieu. While the importance of this locus to *Lm* pathogenicity remains unclear, the observation that this phage-associated protein pair act upon the bacterial cell wall may hold promise in the field of antibiotic potentiation to combat antibiotic resistant bacterial pathogens.

## 1. Importance

Listeriosis is a deadly disease—fatality rates are relatively high even in regions where cases are few and sporadic. Thus, there is still an urgent need to further characterize the pathogenesis of *Lm* and the molecular mechanisms of its virulence. This study broadly characterized a conserved genetic locus that is transcriptionally upregulated during *Lm*’s growth in intracellular conditions. The metzincin and TIMP-like proteins, whose transcription is regulated by this locus, interact in a manner rarely recorded—the TIMP-like lipoprotein is adapted to be cleaved and released to the culture medium by the metzincin. The release of the TIMP-like leads to changes in the bacterial cell morphology and to an increase in bacterial sensitivity to phage lysins and drugs that target the cell wall. This locus may therefore have therapeutic use: to make otherwise resistant bacteria more susceptible to lytic factors and compounds, such as penicillin and phage-encoded lysins, which play an important role in phage therapy.

## 2. Introduction

*Listeria monocytogenes* (*Lm*) is a Gram-positive bacterium, adapted to life as both an environmental saprophyte and an intracellular parasite and pathogen of mammals. If ingested in large quantities, *Lm* may cause Listeriosis, a bacterial infection that is usually mild in healthy individuals, but in severe cases presents as meningitis, encephalitis or a generalized systemic infection [1]. Listeriosis is particularly severe, even lethal, in individuals that are very young, very old, pregnant or immunocompromised, and is still a cause of hundreds of fatalities every year [2].

Bacteriophages (phages) are viruses that infect bacteria, and the most abundant biological entities on Earth [3,4]. Phages can be divided into two groups according to their lifecycle. The first group, with the simpler lifecycle, consists of the lytic phages [5,6]. These phages infect a bacterial host, hijack its replication, transcription and translation machinery to create viral progeny, and finally lyse the host, thereby releasing their progeny. The second group, with the more complex lifecycle, consists of the temperate phages [7,8]. After initial infection of the host, these phages choose one of two lifecycles: they either progress with the infection like a lytic phage, synthesizing progeny and killing the host (termed the lytic cycle), or integrate into the host genome, remain dormant and replicate as a part of the host’s chromosome (termed the lysogenic cycle). Dormant, integrated phage genomes (referred to as prophages) are not trapped in their host, as specific conditions within the host, such as starvation or DNA damage (the latter triggering the SOS response), drive prophages to excise from the chromosome and enter the lytic cycle, a process termed prophage induction.

The principal difference between lytic and lysogenic phages is the presence of a regulatory region in the genome that allows the temperate phage to “choose” either lysis or lysogeny [5,6]. By far the most well-studied case of such a genomic region is the *E. coli* phage λ regulatory switch [7]. This region can bind two mutually inhibiting transcriptional repressors—the λ repressor, encoded by the *cI* gene, and Cro, encoded by *cro*. The λ repressor is responsible for the establishment and maintenance of lysogeny by repressing the expression of *cro* and all subsequent lytic genes. Cro is responsible for repressing the expression of the λ repressor allowing for the expression of the phage’s lytic genes. During phage infection, conditions within the host affect the expression and activity of these repressors, ultimately leading one to overtake the other, determining the lifecycle decision. After lysogeny has been established, *cI* is constitutively expressed so the λ repressor is abundant and active. When the prophage undergoes induction the λ repressor is cleaved, causing it to lose functionality as a repressor. This allows for the transcription of *cro* and the lytic genes, leading the phage to enter the lytic cycle.

Bacteria commonly harbor one or more prophages within their chromosome [9]. Many are infective, meaning they encode all the genes necessary for the synthesis of infective progeny phages and the means to escape from their host (the latter are phage-encoded holins and lysins that trigger bacterial lysis [10]). Others, referred to as cryptic prophages [11], were once infective, but after losing some critical genes, have become trapped in their host chromosome without the ability to produce infective progeny. *Lm* 10403S, the strain used in this study, harbors two prophages: the A118-like infective phage φ10403S [12,13], and a cryptic prophage that encodes for bacteriocins, referred to as the monocin element (short for *L. **mono**cytogenes* bacterio**cin**) [14,15,16].

Research in our laboratory produced two significant discoveries regarding these two prophages and their role in *Lm*’s lifecycle. The first was that φ10403S plays an important role in the ability of *Lm* to infect mammalian cells and replicate within them [12,17]. Briefly, expression of the gene *comK* is important for *Lm*’s ability to infect mammalian cells, but this gene is normally interrupted by the integrated φ10403S prophage. During mammalian infection, φ10403S is induced, excising from the *comK* gene, allowing it to be expressed. Then, following successful infection by *Lm*, instead of completing the lytic cycle, the extrachromosomal circular phage DNA reintegrates into the host genome, returning to the lysogenic cycle. We termed this partial induction “active lysogeny” [18]. The second discovery was that although φ10403S is infective, it is not autonomous—it does not encode its own means to cleave its analogue of the λ repressor [19] (termed the CI-like repressor). Instead, φ10403S is wholly reliant on a monocin-encoded metalloprotease to cleave its CI-like repressor. This effectively reins in its ability to undergo induction at any level in conditions that do not induce the monocin element.

Following these discoveries, we speculated that additional phage-derived elements (or remnants of such) that exist in the *Lm* 10403S genome may play a role in *Lm*’s complex lifecycle. Searching for such elements, we identified a conserved locus encoding two repressors, similar in sequence and configuration to a CI/Cro regulatory switch of a temperate phage. We establish that this locus indeed acts as a regulatory switch in terms of transcriptional dynamics, and that it controls the expression of two adjacent genes: a secreted protease of the metzincin family and a lipoprotein structurally similar to a metzincin inhibitor of the TIMP family. We demonstrate that expression of the metzincin-TIMP-like protein pair affects the bacterial cell wall in trans, increasing susceptibility to phage mediated lysis and cell wall targeting agents. Our data suggest that this remote effect is mediated by the cleavage of the TIMP-like lipoprotein by the secreted metzincin, thereby releasing it into the culture medium as well.

## 3. Materials and Methods

### 3.1. Bacterial Strains and Media

*Listeria monocytogenes* (*Lm*) strain 10403S, *Lm prfA** and *Lm* Mack861 were obtained from Professor Daniel Portnoy (University of California, Berkeley, CA, USA); *Lm* 10403S was used as the WT strain and as background strain for all genetic manipulations unless stated otherwise. *Lm* 10403S strain cured of φ10403S phage (DPL-4056) was generated by Professor Richard Calendar (University of California, Berkeley, CA, USA) by biological curing. *E. coli* XL-1 Blue (Stratagene, La Jolla, CA, USA) was utilized for vector propagation. *E. coli* SM-10 was utilized for conjugative plasmid delivery to *Lm* bacteria. Overnight (O.N.) cultures were grown in BHI with agitation to stationary phase at 37 °C unless stated otherwise. LB MOPS + glucose-1-phosphate (G1P), as described previously [20] and low branch chain amino acid minimal medium (LBMM; MDM with 10 µg/mL BCAAs, as described previously [21]) were prepared as described previously. Conditioned medium (CM) was generated by diluting an O.N. culture to an OD_600_ of 0.03, incubating it at 37 °C with agitation until it reached an OD_600_ of 0.5–0.6 and then filtering the culture using a 0.22 μm filter. *E. coli* strains were grown in Luria–Bertani (LB) (Acumedia, Lansing, MI, USA) medium at 37 °C. Phusion polymerase was used for all cloning purposes and Taq polymerase for verifications of the different plasmids and strains by PCR. Antibiotics were used as follows: chloramphenicol (Cm), 10 µg/mL; streptomycin (Strep), 100 µg/mL; mitomycin C (MC) (Sigma, St. Louis, MO, USA), 1.5 µg/mL. All restriction enzymes were purchased from New England BioLabs, Ipswich, MA, USA.

### 3.2. Generation of Gene Deletion Mutants and Ectopic Expression Strains

To prepare gene deletion mutants, upstream and downstream regions of selected genes were amplified using Phusion DNA polymerase and cloned into pKSV7oriT vector [22]. Cloned plasmids were sequenced and conjugated to *Lm* using the *E. coli* SM-10 strain. *Lm* conjugants were then grown at 41 °C for two days in BHI supplemented with chloramphenicol to promote plasmid integration into the bacterial chromosome by homologous recombination. For plasmid curing, bacteria were passed several times in fresh BHI without chloramphenicol at 30 °C to allow plasmid excision via the generation of an in-frame deletion. The bacteria were then seeded on BHI plates and chloramphenicol sensitive colonies were picked for validation of gene deletion using PCR. The ectopic expression strains were generated by using the pPL2 integrative plasmid to introduce a copy of the relevant gene in trans under the control of a p*_tetR_* promoter [23]. The promoter’s native transcription levels were utilized in this study; no TetR inducer was added.

### 3.3. Lm Growth Assays

O.N. *Lm* cultures were diluted to an OD_600_ of 0.15 (for MC experiments) or 0.03 (for penicillin experiments) and pipetted in triplicates into a 96-well plate, with or without MC (1.5 µg/mL) or penicillin (0.08 μg/mL), bacitracin (250 μg/mL) or D-cycloserine (12 μg/mL). The plates were incubated at 37 °C in a Tecan Infinite M900 PRO plate reader, and the OD_600_ was measured every 15 min proceeded by 2 min of shaking. Cultures were diluted in BHI for MC lysis curve experiments, and either in BHI or CM for penicillin lysis curve experiments.

### 3.4. Mutanolysin and Triton X-100 Lysis Assays

Modified from Popowska et al. (2009) [24]: O.N. bacterial cultures grown at 37 °C with agitation in BHI broth were diluted 1:20 and grown in the same conditions to an OD_600_ of 0.8. The cultures were then centrifuged, washed twice with PBS and resuspended in prewarmed PBS or mutanolysin buffer (50 mM MES, pH = 5.9, 1 mM MgCl_2_) for Triton X-100 and mutanolysin lysis assays, respectively, prewarmed to 37 °C. Triton X-100 (0.1%) or mutanolysin (50 units) were added to the cultures and they were pipetted in triplicates into a 96-well plate and incubated at 37 °C in a Tecan infinite M900 PRO plate reader, and lysis was monitored at OD_600_. Lysis percentage at 2 h post treatment was normalized to untreated culture.

### 3.5. Plaque Forming Assay

Bacteria were grown O.N. at 37 °C with agitation in BHI broth, then the culture was diluted by factor of 10 in fresh BHI, incubated without agitation at 37 °C to reach OD_600_ of 0.4, diluted to an OD_600_ of 0.15, and then a lytic cycle was induced by the addition of MC (1.5 µg/mL) and incubation for 6 h at 30 °C. Bacterial cultures were filtered through 0.22 μm filters that do not allow the passage of bacteria. An appropriate dilution of the filtrates (100 μL) was added to 3 mL of melted LB-0.7% agar medium at 56 °C supplemented with 10 mM CaCl_2_, and 300 μL of an O.N. culture of *Lm* Mack861 bacteria, used as an indicator strain, and quickly overlaid on BHI-agar plates. Plates were incubated for 3–4 days at room temperature to allow plaques to form.

### 3.6. Quantitative Real-Time PCR Analysis

Total nucleic acids were isolated by standard phenol-chloroform extraction methods where 0.04 ng of total nucleic acids were used for analysis of *attB* levels by RT-qPCR using bacterial *rpoD* gene as a reference for sample normalization. For gene expression analysis, the samples were treated with DnaseI, and 1 μg of RNA was reverse transcribed to cDNA using a qScript (Quanta, Beverly, MA, USA) kit. RT-qPCR was performed on 10 ng of cDNA. The relative expression of bacterial genes was determined by a comparison of their transcript levels with those of the bacterial *rpoD* gene, which served as a reference. For transcript level quantification, the expression levels of bacterial genes were compared to a standard curve of 16 to 50,000 copies of genomic *Lm* DNA and normalized to 10,000 copies of *rpoD* per sample. All RT-qPCR analyses were performed using FastStart Universal SYBR Green Master Mix (Roche) on the StepOnePlus RT-PCR system (Applied Biosystems, Waltham, MA, USA) by the standard ΔΔCt method. Statistical analysis was performed using StepOne V2.1 software C_T_ comparative studies upon independent experiments. Error bars represent the 95% confidence interval.

### 3.7. Lm Intracellular Growth

To assess the intracellular growth of *Lm*, 1.5 × 10^6^ Caco-2 or J774 cells were seeded in a 60 mm Petri dish on glass coverslips in 5 mL of Caco-2 growth medium + pen-strep (MEM (Gibco) supplemented with 20% heat-inactivated FBS, 2 mM L-glutamine, 1 mM sodium pyruvate, 1% MEM Eagle nonessential amino acid solution and 1% pen-strep solution; 10,000 units/mL penicillin G sodium salt + 10 mg/mL streptomycin sulfate) or J774 growth medium + pen-strep (DMEM (Gibco) supplemented with 10% FBS, 2 mM L-glutamine and 1% pen-strep solution), respectively, and incubated for 24 h in a 37 °C, 5% CO_2_ forced-air incubator. *Lm* cultures were grown O.N. at 30 °C without agitation and 8 × 10^6^ PBS-washed bacteria were used to infect the cells (MOI of 1). Thirty minutes post infection, the cell monolayers were washed with PBS and fresh medium (no pen-strep) was added. Gentamicin was added 1 h.p.i. to a final concentration of 5 μg/mL in order to limit the growth of extracellular bacteria. At each time point, three coverslips were transferred into 5 mL of sterile water to release the intracellular bacteria. Serial dilutions of the resulting lysate were plated on BHI agar plates and the CFUs were counted after 24 h incubation at 37 °C.

### 3.8. Transcription Analysis of Intracellular Bacteria

Modified from Sigal et al. (2016) [25]: 0.5 mL of O.N. culture of WT *Lm* bacteria grown at 30 °C was washed twice in PBS and used to infect ~20 × 10^6^ J774 cells seeded in a 145 mm dish, resulting in a multiplicity of infection (MOI) of ~100. Thirty minutes post infection, J774 monolayers were washed twice with PBS to remove unattached bacteria and fresh medium was added. At 1 h.p.i., gentamicin (50 µg/mL) was added to limit extracellular bacterial growth, then 6 h post infection, intracellular bacteria were liberated from the macrophages by washing with 20 mL ice cold H_2_O (RNAse free-DEPC treated), collected by passing the lysate through 0.45 μM filter membranes and flash-frozen in liquid nitrogen. Bacteria were recovered from the filters by vortexing into AE buffer (50 mM NaOAc pH 5.2, 10 mM EDTA), and bacterial nucleic acids were extracted using hot (65 °C) phenol with 1% SDS followed by ethanol precipitation. The samples were treated with DnaseI, and the resulting of RNA was reverse transcribed to cDNA using a qScript (Quanta, Beverly, MA, USA) kit. The relative expression of bacterial genes was determined by a comparison of their transcript levels with those of the bacterial *rpoD* gene, which served as a reference. All RT-qPCR analyses were performed using FastStart Universal SYBR Green Master Mix (Roche, Basel, Switzerland) on the StepOnePlus RT-PCR system (Applied Biosystems, Waltham, MA, USA) by the standard ΔΔCt method. Statistical analysis was performed using StepOne V2.1 software C_T_ comparative studies upon independent experiments. Error bars represent the 95% confidence interval.

### 3.9. Biofilm Formation Analysis

Overnight *Lm* cultures, grown in BHI at 30 °C without agitation were diluted in BHI or CM to an OD_600_ of 0.05 and pipetted in triplicates into a 96-well plate. The plate was incubated for 24 or 48 h at 37 °C without agitation. The cultures were then gently vacuumed from their wells, the wells were gently washed twice with PBS and the plate was dried briefly upside-down on a paper towel. One hundred microliters 100% ethanol was added to each well, and after 1 min at room temperature, the ethanol was discarded and the plates dried on a paper towel. Two hundred microliters crystal violet solution (0.4% crystal violet, 12% ethanol) was added to each well, and following incubation at room temperature for 2 min, the solution was discarded, the wells were washed three times under running water and then dried on a paper towel. One hundred microliters 1% SDS solution in PBS was added to each well, pipetted up and down briefly and the plate was gently stirred on a horizontal shaker for 10 min. The content of the wells was transferred to a new 96-well plate and OD_595_ was measured in a Synergy HT BioTek plate reader.

### 3.10. Western Blot Analysis

O.N. cultures of *Lm* strains harboring 6His-tagged proteins under the regulation of a p*_tetR_* promoter in the integrative pPL2 plasmid were diluted × 100 and grown at 37 °C in 50 mL BHI to an OD_600_ of 0.5–0.6. The bacteria were then collected by centrifugation and the supernatant was used as the secreted fraction. After collecting the supernatant, the bacteria were washed with 1 mL prechilled Buffer A (20 mM Tris-HCl pH 8, 0.5 M NaCl and 1 mM EDTA) and transferred to microtubes. The suspension was then pelleted and washed with mutanolysin buffer, resuspended in 1 mL mutanolysin buffer, treated with 25 units/mL of mutanolysin (Sigma, St. Louis, MO, USA) and incubated at 37 °C for 1 h. The treated bacteria were then centrifuged for 3 min at 3000× *g*, the supernatant was gently removed, the pellet was resuspended in cold Buffer A supplemented with 1 mM PMSF and was lysed by ultrasonication. Cell debris was removed by centrifugation at 3000× *g* and the supernatant was further centrifugated at 100,000× *g* for 15 min (4 °C). The supernatant was used as the cytosolic fraction and the pellet, solubilized by 1% SDS, was used as the membrane fraction. Cytosolic and total secreted proteins (supernatants) were precipitated by trichloroacetic acid (TCA). Total protein content in each sample was quantified by a modified Lowry method and samples with equal amounts of total proteins were separated on 12.5% SDS-polyacrylamide gels and transferred to nitrocellulose membranes. Proteins were probed with rabbit anti-6His tag antibody (Abcam ab9108) at 1:1000 dilution, followed by HRP-conjugated goat antirabbit IgG (Jackson ImmunoResearch, West Grove, PA, USA) at 1:20,000 dilution. Western blots were developed by a homemade enhanced chemiluminescence reaction (ECL). Images were obtained using Amersham Imager 800 (GE Healthcare Life Sciences, Chicago, IL, USA).

### 3.11. Protein Sequencing of Tim

*Lm* harboring a 6His tagged Tim coexpressed with an untagged Mtz, under the regulation of a p*_tetR_* promoter on the integrative pPL2 plasmid was grown at 37 °C in 50 mL BHI to an OD_600_ of 0.5–0.6. The culture was then filtered using a 0.22 μm filter and the filtrate was incubated with 0.5 mL Ni-NTA beads (Takara, Kusatsu, Shiga, Japan) for 3 h at 4 °C with tilting. The Ni-NTA beads were then loaded on a column and washed with 10 mL of Buffer-P (0.3 M NaCl, 50 mM NaH_2_PO_4_, pH = 8) supplemented with 10 mM imidazole and then with 25 mM imidazole. The protein was eluted by 250 mM of imidazole in Buffer-P and precipitated by TCA. The proteins were separated on 12.5% SDS-polyacrylamide gels and stained with Coomassie brilliant blue to yield two primary visible bands: the full-length TIMP-like and its cleavage product. The stained band representing the cleavage product was isolated from the gel and analyzed by peptide mass fingerprinting at The Smoler Protein Research Center at the Technion, Haifa, Israel. Protein samples were digested by chymotrypsin, and the resulting proteolytic peptides were analyzed by LC–MS/MS on Q-Exactive (Thermo, Waltham, MA, USA) and identified by Discoverer software version 1.4.

### 3.12. Transmission Electron Microscopy

Following 4 h of growth in BHI at 37 °C with or without MC (1.5 µg/mL), bacteria were collected by centrifugation and fixed in 2.5% glutaraldehyde in PBS O.N. at 4 °C. After several washings in PBS, the bacteria were postfixed in 1% OsO_4_ in PBS for 2 h at 4 °C. Dehydration was carried out in graded ethanol followed by embedding in glycidether. Thin sections were mounted on Formvar/Carbon coated grids, stained with uranyl acetate and lead citrate, and examined by a Jeol 1400–Plus transmission electron microscope (Jeol, Tokio, Japan). Images were captured using the SIS Megaview III and iTEM TEM imaging platform (Olympus, Tokyo, Japan).

### 3.13. Scanning Electron Microscopy

Following 4 h of growth in BHI at 37 °C with or without MC (1.5 µg/mL), bacteria were collected by centrifugation, washed three times with PBS and fixed in 2.5% glutaraldehyde in PBS. After washing with PBS, the samples were postfixed in 2% OsO_4_ solution for 2 h at 4 °C and dehydrated by successive ethanol treatment. After critical point drying using Balzer’s critical point dryer, the samples were mounted on aluminum stubs and sputter-coated (SC7620, Quorum) with gold. Images were captured on the scanning electron microscope (SEM) (JCM-6000, Jeol, Tokio, Japan).

## 4. Results

### 4.1. Listeria monocytogenes Harbors a Conserved Phage Regulatory Switch-Like Locus within Its Chromosome

Among the products of our search for additional phage-derived elements or phage remnants in *Lm* 10403S was a genetic locus encoding two putative opposite-facing XRE-family transcriptional regulators (LMRG_00711-00712), highly reminiscent of a temperate phage *cI-cro* regulatory switch (Figure 1A). Two additional genes were found adjacent to *cI-cro*-like pair: the first, *LMRG_00713*, encoding a protein with predicted Sec/SPI signal peptide and an extended zinc metalloproteinase catalytic site motif (**HE**XX**H**XX**G**XX**H/D**) followed by a Met-turn motif (AI**M**LD), making it a putative member of the metzincin metallo-endopeptidase superfamily [26]. Metzincins can be found in a myriad of single- and multicellular organisms, are commonly secreted to the medium or externally bound, and their activity is usually tightly regulated via the presence of an N-terminal inhibitory domain or an endogenous inhibitor.

The second gene, *LMRG_00714*, is located immediately after *LMRG_00713* with no apparent promoter. It harbors a predicted Sec/SPII signal peptide, including a lipobox motif [27] (LTAC) in the signal peptide C-terminus, making it a putative lipoprotein, and contains a region similar in structure (but not in sequence) to a tissue inhibitor of metalloproteinases (TIMP; predicted by Phyre2 [28], ~60% confidence). Coincidentally, TIMPs are proteins that act upon matrixins, a family of metzincins common in vertebrates involved in the remodeling and repair of tissue, usually as inhibitors but occasionally as activators [26].

Phylogenetic analysis of these four genes reveals that each pair of genes, i.e., the *cI-cro*-like and metzincin-TIMP-like (henceforth referred to as *mtz* and *tim*), bear distinct histories. The *cI-cro*-like pair is exceptionally well conserved in all currently recognized species of the *Listeria* genus. They can be found proximal to *LMRG_00715* homologues and genes encoding a homologue of the trigger factor chaperone (*tig*) and the ClpX component of the Clp protease (*clpX*) on one end, and genes encoding a PrsW-like protease and components of the proline biosynthetic pathway (ProA and ProB) on the other end in nearly all members of the *Listeria* genus (Figure 1B). Genes with high sequence similarities to the *cI-*like (*LMRG_00711*) (HipB superfamily transcriptional regulators, ~60% similarity, not including the HTH domain) can also be found in other Firmicutes species related to *Listeria*, but unlike their *Listeria* counterpart, they frequently lack the additional XRE protein and are usually in the vicinity of a gene encoding an ImmA antirepressor, which belongs to the M78-family metallopeptidases (Appendix A). The *mtz-tim* gene pair are also conserved within the genus *Listeria*, but are limited to the *Listeria* sensu strictu clade [29], which includes the two known pathogenic species of *Listeria* (*L. monocytogenes* and *L. ivanovii*) and whose common ancestor is believed to have been a pathogen [30]. Their location within the genome is also fairly conserved among members of this clade with some variability, but always proximal to genes of phage origin or within prophages (Figure 1C). In these prophages, the *mtz-tim* pair are accessory genes, frequently found downstream of the phage holin-lysin and upstream of another pair of genes, homologues of which are also present in φ10403S (*LMRG_01558-1559*). They share high sequence similarity with *Lm* 10403S *mtz-tim* pair, and their putative promoter regions contain a long DNA sequence identical to that upstream the *agrBDCA* operon [31], indicating they are most likely regulated by the Agr quorum-sensing system. (Figure 1C). The association of the *mtz-tim* gene pair with mobile genetic elements is also evident in other closely related Firmicutes (e.g., species of *Bacillus*, *Brevibacillus*, *Lysinibacillus* and *Clostridium*), where similar gene pairs are consistently found proximal to genes encoding integrases, transposases, recombinases and other genes commonly found in phage and integrative conjugational elements (Appendix A). Interestingly, in these closely related Firmicutes, the homologue of the *tim* have no lipobox and harbor a SPI signal peptide instead of a SPII signal peptide, indicating they are unanchored secretory proteins and not lipoproteins.

In summary, these two adjacent pairs of genes—*cro-cI*-like XRE family transcriptional regulators and *mtz-tim*—are conserved among the pathogenic clade of the *Listeria* genus and are likely of phage or mobile genetic element origin.

### 4.2. The cI-cro-Like Locus Possesses the Transcriptional Dynamics of a Regulatory Switch

As previously stated, two principal traits of a *cro-cI* regulatory switch are (1) it encodes two mutually inhibiting repressors, and (2) it undergoes induction under specific conditions. To test whether the *cI*- and *cro*-like genes of this locus repress each other’s gene expression, we used real-time quantitative PCR (RT-qPCR) to measure the transcription levels of the locus’ four genes in strains deleted for either the *cI*-like or the *cro*-like (*Lm* Δ*cI*-like and *Lm* Δ*cro*-like, respectively), or expressing the *cro*-like using the integrative pPL2 plasmid under the regulation of a p*_tetR_* promoter (pPL2-*cro*-like), during mid-logarithmic growth in BHI at 37 °C in comparison to wild type *Lm* (WT *Lm*). In line with our hypothesis, deletion of the *cI*-like induced the expression of the *cro*-like and the subsequent *mtz-tim* genes (Figure 2A), whereas deletion of the *cro*-like had no effect on gene transcription (Figure 2B). Ectopic expression of *cro*-like repressed the expression of *cI*-like, resulting in the induction of *mtz-tim* (Figure 2C). These results demonstrate that the two regulators encoded in this locus are mutually inhibiting repressors, resembling the classic CI and Cro phage regulators.

To test which conditions, if any, lead to the induction of this regulatory switch locus, we quantified transcription levels of its four genes in WT *Lm* grown to mid-log in BHI and 37 °C and then compared them to the levels measured in a variety of other conditions and strains. Notably, the locus was repressed in the mid-log BHI condition (37 °C), with *cI*-like being highly transcribed and *cro*-like, *mtz* and *tim* genes far less so (Figure 3A). It was also repressed in bacteria grown to the logarithmic phase in BHI at 30 °C, in LB-MOPS supplemented with glucose-1-phosphate (a rich medium that mimics the intracellular niche [20]) at 37 °C, and in a *Lm* strain expressing a constitutively active *prfA* (*prfA**), the master regulator of *Lm*’s virulence genes [32], grown in BHI at 37 °C (Figure 3B). Surprisingly, the locus remained repressed in bacteria treated with Mitomycin C (MC), a compound that causes DNA damage which activates the bacterial SOS response, subsequently inducing prophages and additional mobile genetic elements (Figure 3B). Instead, expression of *cro*-like, *mtz* and *tim* was induced in bacteria grown to the stationary phase in BHI at both 30 and 37 °C, in bacteria grown in low branched-chain amino acid minimal defined medium (LBMM, a defined medium that mimics the conditions of the intracellular niche [21,33]) (Figure 3B) at 37 °C, and in bacteria grown intracellularly in the J774 macrophage cells (Figure 3C).

To summarize, these data demonstrate that similar to phage regulatory switches, this locus consists of two mutually inhibiting repressors, CI-like and Cro-like, and undergoes induction in a number of conditions, namely upon growth to stationary phase, BCAA starvation and within mammalian cells.

### 4.3. Mtz and Tim Localize to the Membrane and to the Culture Medium, Where Mtz Likely Cleaves Tim

Having established the conditions in which the *cro*-*cI*-like regulatory switch is induced, we proceeded to characterize the genes regulated by it—*mtz* and *tim*. We began their characterization by experimentally determining the localization of their product proteins. The observation that these two genes nearly always come in pairs in *Listeria* and other closely related Gram-positive bacteria lead us to suspect that they may interact or affect each other’s localization, and so we assessed the localization not only of each protein individually, but also while coexpressed with its partner. Thus, we cultured *Lm* strains ectopically expressing C-terminal 6His tagged Mtz or Tim using p_tetR_ promoter, either alone or coexpressed with an untagged cognate protein (using pPL2-*mtz*-his, pPL2-*tim*-his, pPL2-*mtz*-his-*tim* and pPL2-*mtz*-*tim*-his, respectively), in BHI at 37 °C. The cell culture of each strain was then fractionized to cytosol, membrane and supernatant, and each fraction was subjected to Western blot analysis using anti-6His antibodies.

Both proteins were predicted to harbor an N-terminal signal peptide and Tim was further predicted to harbor a lipobox, and so we anticipated that Mtz would be found mainly in the culture supernatant and Tim in the membrane fraction. Both predictions were confirmed following fractionation and precipitation of proteins from *Lm* strains expressing either a tagged Mtz or Tim, though a substantial amount of Tim was also found in the culture supernatant (Figure 4A–C). In addition to the full-length Tim, the supernatant of *Lm* ectopically expressing a tagged Tim contained additional shorter variants of Tim, indicating this protein undergoes proteolytic cleavage (two cleavage products are marked with asterisks, Figure 4C). When a tagged Tim was coexpressed with Mtz, a single short variant was observed, suggesting that Mtz mediates a specific cleavage of Tim (Figure 4C). To examine whether the protease activity of Mtz is necessary for Tim cleavage, we generated an *Lm* strain coexpressing a 6His-tagged *tim* with an inactivated variant of Mtz, harboring a E136A mutation in its HEXXH catalytic site motif [34,35] (pPL2-*mtz*(inactive)-*tim*-his) and subjected its culture supernatant to a Western blot assay. Remarkably, the Western blot pattern of Tim coexpressed with the inactive Mtz was identical to that of Tim expressed alone (demonstrating two nonspecific cleavage products) (Figure 4C,D), giving further proof that Mtz specifically cleaves Tim.

We next separated the cleavage product of the tagged Tim, obtained from the culture medium of *Lm* coexpressing Mtz and a tagged Tim, and subjected it to mass spectrometry analysis and found it was missing ~50 N-terminal amino acids (Figure 4E). Notably, the predicted cleavage region (~amino acid 49 to 55) appears to be conserved among the Tim homologues of *Listeria* species, constituting a possible cleavage site consensus sequence. Furthermore, structural analysis of the cleaved Tim in Phyre2 increased the confidence of the TIMP domain prediction to over 90% (compared to ~60% of the full-length Tim protein).

In conclusion, these results indicate that Tim localizes to the bacterial membrane and the cell culture, and that a fraction of Tim is cleaved by the secreted Mtz, yielding a short soluble Tim protein.

### 4.4. Expression of Both Mtz and Tim Sensitize the Bacteria to Phage Mediated Bacterial Lysis

Given the locus’ association with prophages and mobile genetic elements, we investigated whether it, or more specifically the *mtz-tim* pair, affect *Lm* phage elements, i.e., φ10403S and the monocin element. We did so by monitoring and comparing the growth of WT *Lm* and *Lm* strains expressing *mtz* and *tim* either alone or together (using pPL2-*mtz,* pPL2-*tim* and pPL2-*mtz*-*tim*)*,* with and without MC treatment (to induce the phage elements) at 30 or 37 °C. Notably, we previously established that a temperature of 37 °C is less conducive to the lytic induction of φ10403S than 30 °C [17]. All strains exhibited a similar growth pattern without MC at both temperatures, and a similar culture-wide lysis patterns when grown at 30 °C with MC (Appendix A). In line with our previous findings, when WT *Lm* was grown with MC at 37 °C, instead of demonstrating culture-wide lysis, the culture’s growth was merely arrested (Figure 5A). Similarly, *Lm* expressing either Mtz or Tim individually, or Tim coexpressed with a catalytically inactive Mtz (using pPL2-*mtz*(inactive)-*tim*) also exhibited growth arrest when grown in these conditions (Figure 5A,B and Appendix A). However, when bacteria expressing both Mtz and Tim were grown with MC at 37 °C, a culture-wide lysis pattern was observed (Figure 5A). These findings suggest that when both Mtz and Tim are expressed, their proteolytic interaction leads to lysis of *Lm* in conditions where phage lysis is normally hampered.

To assess whether the source of this lysis is indeed the phage elements and not the Mtz-Tim pair itself, we conducted a similar experiment, this time expressing the pair in an *Lm* strain cured of its two phage elements (Δ*phages*, deleted for φ10403S and the monocin element). No lysis was observed in the phage cured bacteria, implicating the phage elements as the lysing agents (Figure 5C), suggesting that Mtz-Tim bolster phage mediated lysis. Moreover, when *Lm* bacteria expressed the *mtz-tim* pair under these same conditions (i.e., growth in BHI at 37 °C with MC), the number of φ10403S virions released into the medium was greater than that of WT *Lm* (~5-fold) (Figure 5D), further associating Mtz and Tim with the phage lytic pathway.

To examine whether the enhanced lysis and virion production observed in strains expressing *mtz* and *tim* is caused by increased rates of phage induction, we measured phage excision and lytic gene expression with and without MC treatment using RT-qPCR. Phage excision can be deduced by comparing the number of intact bacterial attachment sites and excised phage genomes (*attB* and *attP,* respectively). To assess lytic gene expression, we monitored the expression of two early lytic genes and two late lytic genes from both φ10403S and the monocin element. Interestingly, analysis of these induction markers established no difference between WT *Lm* and *Lm* expressing *mtz-tim* grown in BHI at 37 °C with MC (Figure 5E,F), indicating no difference in phage induction in these conditions.

Taken together, these results demonstrate that expression of Mtz and Tim does not affect phage induction, but rather enhances the sensitivity of the bacteria to phage mediated lysis, which in this case is driven by the coordinated action of the phage-element lysis proteins, holin and lysin. Holins tend to oligomerize within the bacterial membrane and form holes, either allowing the lysins to pass through the membrane to the cell wall or activating them via membrane potential collapse. The lysins then degrade the cell wall peptidoglycan, compromising its integrity and leading to cell lysis [36].

### 4.5. Expression of Mtz and Tim Changes Lm Cell Morphology

In order to unravel the observed enhanced sensitivity to phage lysis, we further characterized the *Lm* strain ectopically expressing *mtz-tim*. While observing this strain (grown to mid-log in BHI at 37 °C) under a scanning electron microscope (SEM), we noticed a morphological difference compared to WT *Lm*: WT bacteria presented as completely straight rods, yet ~40% of the bacteria expressing *mtz-tim* exhibited a curved or folded morphology (Figure 6A, upper panel). Treating the bacteria with MC, which caused them to elongate as a part of the DNA damage response, made the curved phenotype far more evident (Figure 6A, lower panel). We next observed thin sections WT *Lm* and *Lm* expressing *mtz-tim* under a transmission electron microscope (TEM). While we could not discern between the two strains when they were grown without MC treatment, with the addition of MC, not only was the previous curved morphology replicated in the *mtz*-*tim* expressing strain, but we also observed a thick layer of peptidoglycan at the curves and what seemed to be asymmetric fission (the latter can be also a result of the curved morphology) (Figure 6B).

Of note, similar curved morphologies were observed in past studies of *Lm* penicillin binding proteins (PBPs) [37,38]. In these studies, PBPs were shown to mediate *Lm*’s resistance to a variety of external stresses and affect cell wall composition and rigidity. Incidentally, changes in the cell wall could conceivably enhance or attenuate the action of lytic factors, such as phage-encoded lysins. Thus, we arrived at the hypothesis that the Mtz-Tim pair acts upon the cell wall, thereby changing *Lm* morphology and increasing its susceptibility to phage lysis.

### 4.6. Secreted Mtz and Tim Increase Bacterial Susceptibility to Cell Wall Targeting Antibiotics, Also in Neighboring Cells

Under this new hypothesis, if the Mtz-Tim pair acts upon the cell wall in a manner that sensitizes *Lm* to lysis, we assumed that it would enhance the effects of lysis-inducing factors other than phage holin/lysin as well. The first compound we decided to test was penicillin G. The antibiotic effect of penicillin is based upon its capacity to permanently bind DD-transpeptidases, inhibiting the formation of new peptidoglycan crosslinks, hence weakening the integrity of the cell wall [39]. We next evaluated the effects of *mtz*-*tim* expression (expressed together or individually) on the growth of WT *Lm* in BHI at 37 °C with and without a sublethal concentration of penicillin G. In line with our hypothesis, growth of *Lm* coexpressing *mtz*-*tim* was inhibited by penicillin to a greater degree than that of WT *Lm* (Figure 7A). Similar results were obtained in a strain of *Lm* lacking the two phage elements (Δ*phages*) (Appendix A). Akin to the lysis phenotype presented earlier, the bolstered growth-inhibiting effects of penicillin were dependent upon the expression of both Mtz and Tim (Figure 7B) and the proteolytic activity of Mtz (Figure 7C). Notably, examining other cell-wall targeting antibiotics, such as, bacitracin and D-cycloserine, we observed a similar growth inhibition of bacteria overexpressing *mtz*-*tim* (sublethal concentrations were used, Appendix A). We next examined the effect of *mtz-tim* expression on the effectiveness of other lysis-inducing agents, such as, mutanolysin and Triton X-100. Adding these agents to bacterial cultures of WT *Lm* or *Lm* ectopically expressing *mtz-tim* in the late-log growth phase (OD = 0.8), we observed that under both treatments *mtz*-*tim* expression bolstered bacterial lysis (Appendix A).

The apparent interaction of the Mtz and Tim and the presence of the cleaved variant of Tim in the culture medium led us to hypothesize that the Mtz-Tim pair does not only affect the bacteria that originally express them, but their neighbors as well. To test this, we conducted a similar growth experiment, this time using conditioned media (CM) that was previously inhabited by WT *Lm* or strains of *Lm* ectopically expressing *mtz* (either active or inactive) and *tim*. Remarkably, penicillin’s growth inhibiting effects upon WT *Lm* were greatly enhanced when it was grown in media previously inhabited by *Lm* expressing both *tim* and a catalytically active *mtz* (Figure 7D). These findings indicated that the factors that increase lysis susceptibility require the activity of Mtz and freely diffuse within the culture medium, remotely acting upon neighboring cells. To further confirm that Mtz and Tim themselves sensitize the bacteria to penicillin, and not other factors such as small quorum-sensing pheromones (some of which are heat stable peptides) [40], we conducted a similar penicillin-treated CM growth experiment, but either boiled the CM or depleted it of all proteins with a molecular weight higher than 3 kDa using size-exclusion centrifugal filters (Figure 7E,F). In both cases, the treated conditioned medium lost its lysis-enhancing properties, thereby excluding the possibility that quorum-sensing molecules or pheromones mediate this phenotype. In light of these findings, we can speculate that Mtz and Tim themselves sensitize *Lm* to lysis when secreted into the culture medium.

### 4.7. Expression of Mtz and Tim Hinders Lm Invasion of Mammalian Cells and Biofilm Formation

Having characterized the effects of *mtz-tim* expression upon *Lm*, we proceeded to examine its impact on key stages of *Lm*’s complex lifecycle. We examined two fundamental processes: biofilm formation and infection of mammalian cells. Interestingly, a previous study that screened for genes important for biofilm formation in *Lm* established that insertion of a mariner transposon in the intergenic region between the *cI*-like and *cro*-like genes of this locus inhibits biofilm formation after 24 and 48 h of growth in BHI at 37 °C [41]. We hypothesized that this effect was caused by the overexpression of the *mtz*-*tim* pair as a result of impaired CI-like repression. Accordingly, we conducted a biofilm formation assay on WT *Lm* and *Lm* strains deleted for *cI-*like (*Lm* Δ*cI-*like) and its complemented strain (*Lm* Δ*cI-*like + pPL2-*cI-*like), deleted for *mtz-tim* (*Lm* Δ*mtz*-*tim*) and ectopically expressing *mtz-tim* (using pPL2-*mtz-tim*). The strains were grown in BHI at 37 °C for 24 h without shaking. In line with our hypothesis, both the strain deleted for *cI*-like and the strain expressing *mtz*-*tim* exhibited reduced biofilm formation compared to WT *Lm* in these conditions (Figure 8A). Of note, deletion of the *mtz-tim* pair had no effect on biofilm formation. We next examined whether Mtz and Tim play a role in *Lm* infection of mammalian cells. For this purpose, we used WT *Lm* and *Lm* strains deleted for *mtz-tim*, deleted of the *cI*-like gene and deleted for *cI-*like while ectopically expressing *cI-*like to infect J774 and Caco-2 cells, models of mammalian immune cells and intestinal epithelial cells, respectively. Infected cells were sampled every two hours and quantified for intracellularly grown bacteria. The results demonstrated that *Lm* deleted for *mtz-tim* invades and grows in mammalian cells at the same rates as WT *Lm*, while a mutant deleted for the *cI*-like exhibits lower infection rates of mammalian cells but no effect on intracellular growth (Figure 8B,C). Taken together, we could not prove that native induction of this locus, triggering the expression of *mtz* and *tim*, affects the formation of biofilm or invasion of mammalian cells in our models, but we demonstrated that expression of *mtz* and *tim* can hinder both processes.

## 5. Discussion

Although mobile genetic elements (MGEs) of prokaryotes, such as bacteriophages, are parasites that burden their hosts, sometimes fatally so, they are also facilitators of their evolution and survival in new niches. These elements are completely reliant upon their hosts’ machinery and metabolism for their continued existence and so, as they mobilize and disseminate in the host population, they face evolutionary pressures to minimize harm to their hosts and even to aid them when possible [42]. Deleterious genes in bacterial genomes are subject to intense evolutionary pressures, precipitating their rapid deletion [43,44]. Unfortunately for MGEs, particularly integrative elements (e.g., temperate phages and ICEs), this leads to the reduction and rearrangement of their genomes, sometimes to the point of a complete loss of autonomy and original function. Accordingly, previous studies suggest that many, if not most integrated phages are defective in some manner [19,45,46]. In this process, termed domestication, the defective MGEs retain only those genes useful for their host. Surprisingly, these include core MGE genes such as of phage tail proteins, which have evolved to become secretion systems and bacteriocins [15,47,48], and transcriptional regulators, whose regulatory specificity and/or dynamics changed to aid their host [49,50]. Such may be the case in the *cI-cro/mtz-tim* locus characterized in this study.

The intricate symbiosis between *Lm* and its phage elements and its impact on *Lm*’s complex lifecycle drove us to study an as of yet uncharacterized locus in the *Lm* genome resembling a phage regulatory switch. Phylogenetic observations divided this locus into two distinct and conserved elements: the *cI-cro*-like switch and the *mtz-tim* pair. The fact that *mtz* and *tim* were consistently observed in pairs, not only in other members of the *Listeria* sensu strictu clade but also across a multitude of Gram-positive bacteria (Appendix A), is likely no coincidence: metzincin activity is usually heavily regulated and Tim is structurally highly similar to metzincin inhibitors of the TIMP family. While we could not find any records of bacterial proteins similar in structure to TIMPs in the literature, let alone in the context of metzincin regulation, there are known instances of bacterial metzincins being regulated by endogenous inhibitors, such as the *Pseudomonas aeruginosa* alkaline protease and its inhibitor [51]. The alkaline protease inhibitor is not similar in structure to TIMPs, but the two have been shown to interact with their cognate metzincin in a similar fashion [52].

In this work, we showed that the Mtz-Tim pair has potent effects when ectopically coexpressed. They cause *Lm* (both the bacteria expressing them and their neighbors in the culture medium) to be more sensitive to a variety of lytic agents: phage holin-lysin, penicillin, mutanolysin and Triton X-100. The combination of increased lysis sensitivity and the curved phenotype observed under electron microscopy suggests that Mtz and Tim act upon the bacteria cell wall. As indicated, cell wall deficiencies caused by the deletion of penicillin binding proteins PBPs have been shown to cause similar effects (i.e., sensitivity to penicillin and a curved phenotype) [37,38]. These PBPs contain carboxypeptidase or transpeptidase domains, raising the possibility that Mtz and Tim directly affect these enzymes (which are also located extracellularly) or indirectly modulate the cell wall crosslinking or peptidoglycan turnover.

Our characterization of the interaction between Mtz and Tim started with the observation that variants of Tim, a lipoprotein that should be anchored to the membrane, are also present in the culture medium (Figure 4C). We found that Mtz mediates the cleavage of Tim, yielding a short Tim variant that is highly similar in structure to TIMP proteins. We also found a large amount of the full length Tim in the culture supernatant, an observation that can be explained by the presence of Tim on extracellular membrane vesicles, which have been shown to carry lipoproteins [53,54]. Interestingly, proteolytic cleavage of lipoproteins in bacteria with functional lipoprotein biogenesis machinery is an uncommon and poorly studied phenomenon [55,56,57,58]. Although the identity of the relevant protease is rarely discovered in these studies, our experiments implicated Mtz as the protease that cleaves Tim. All the lysis susceptibility phenotypes observed in *Lm* expressing *mtz-tim* were also dependent upon the proteolytic activity of Mtz (Figure 5 and Figure 7), suggesting the cleavage of Tim by Mtz mediates these phenotypes. Moreover, the lysis-enhancing properties of conditioned media was also dependent upon the expression of both Tim and a catalytically active Mtz (Figure 7D). This associates the proteolytic cleavage of Tim by Mtz to the presence of a freely diffusing lysis-enhancing factor in the conditioned media. Overall, we can suggest the following model for Mtz-Tim interaction—Tim is inactive in its membrane-anchored, mature lipoprotein form. However, when it interacts with the secreted Mtz, it is cleaved, and the TIMP-like fragment is secreted to the culture medium. This fragment, either alone or together with Mtz, is the cause for all phenotypes observed in this work.

While many integrated MGEs are induced when the host’s SOS response is activated [8,59], unexpectedly, the identified *cI-cro/mtz-tim* locus is not (Figure 3B). Close examination of its proximal genes demonstrates that unlike some of its homologues in related bacteria, this locus is missing an endogenous factor that inactivates the CI-like repressor, such as an ImmA-like antirepressor (Appendix A). Since we found this locus to be induced during growth in intracellular or intracellular-like environments and in the stationary phase of growth, it is likely that its transcriptional control has evolved to be adapted to the mammalian environment. We tested the significance of *mtz-tim* expression upon invasion of mammalian cells and biofilm formation, both being crucial for *Lm* pathogenicity. Forming biofilm allows *Lm* to survive in the harsh conditions of food processing and storage plants [60], and its ability to infect and grow within mammalian cells allows it to successfully replicate and spread in human tissues. We expected the absence of *mtz-tim* to have some deleterious effect upon either process, but none was observed in our models (Figure 8A–C). *Lm* infects mammals by invading a variety cell and tissue types, so in this respect, our models are relatively limited. Similarly, *Lm* forms biofilms on many substrates and temperatures, yet we only tested the effect of *mtz-tim* on biofilm formation in one condition—in BHI medium at 37 °C. Thus, conducting additional experiments in a greater variety of cells, media, etc. could shed light on the possible adaptive roles of this locus. We also tested the effects of ectopic *mtz-tim* expression on the ability of *Lm* to invade mammalian cells and to form biofilm. Surprisingly, it obstructed both processes (Figure 8A–C). This demonstrates that tight regulation of *mtz-tim* transcription is key to *Lm* biofilm formation and invasion of mammalian cells, as opposed to intracellular growth or growth in rich media (Appendix A), which are unaffected by *mtz-tim* expression.

The role of Mtz and Tim in *Lm* lifecycle and in virulence, if any, remains unclear. The observation that these proteins are also encoded by various listerial prophages suggests that they contribute to the lytic production of the phages or to their interaction with the host. The location of these genes downstream the holin and lysin genes, in addition to their enhancing effect on phage mediated lysis, suggest that they possess accessory lysis functions that probably play a role under certain environmental conditions found in nature. For example, the prediction that the phage-encoded *mtz-tim* genes are regulated by the Agr system, raise the possibility that they are involved in the regulation of bacterial lysis and virion release at the population level. In the case of the identified Mtz-Tim pair, it is possible that these proteins were domesticated to aid *Lm* in the mammalian environment in a manner that is independent of the phage elements. We can speculate regarding their possible function based on studies of similar proteins in other bacteria. One possibility is that Mtz cleaves the proteins of either the bacteria or its mammalian host, allowing *Lm* to evade the host’s innate and/or adaptive immune system, with Tim acting solely as a regulator of Mtz. Such a mechanism can be seen in *Pseudomonas* bacteria, with their alkaline protease, a secreted metzincin and its periplasmic inhibitor AprI [61,62,63]. Another possibility is that Mtz’s role is only to cleave Tim, and Tim’s released fragment acts as a virulence factor, modulating host cellular or innate-immunity activities. A similar mechanism has been recorded in *Mycoplasma fermentas*, but in this case, after the cleavage of the lipoprotein MALP-404 occurs, it is the anchored fragment and not the released fragment that modulates the activity of host innate immune system [55].

Finally, the penicillin-potentiating effects of Tim and Mtz may be of great therapeutic value. Although *Lm* infections are currently treatable with β-lactam antibiotics [64], this treatment method may not last as β-lactam resistance is common in other pathogenic bacteria [65,66]. Antibiotic resistance has naturally led to research exploring factors and therapeutic approaches that counter this resistance [67,68,69,70]. Further characterization of these proteins in *Lm* and other pathogenic bacteria may be of aid in the fight against bacterial resistance to β-lactams and other antibiotics that attack the bacterial cell wall.

## Figures and Tables

**Figure 1 microorganisms-09-01323-f001:**
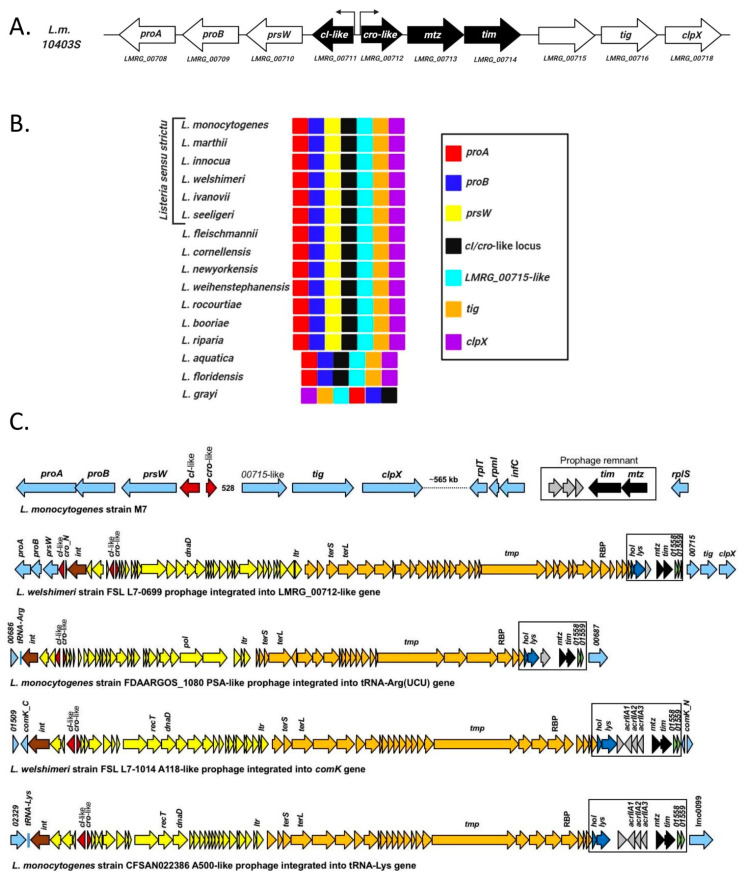
Bioinformatic analysis of the *cI-cro-*like locus. (**A**) Schematic representation of the *cI-cro-*like locus and proximal genes in *Lm* strain 10403S. (**B**) The *cI-cro-*like locus and its proximal genes are highly conserved in the *Listeria* genus, with slight variation only in *L. grayi*. Genes are presented in color code as indicated. (**C**) Examples of different locations of the metzincin-TIMP-like pair (*mtz-tim*) in other members of the *Listeria* sensu strictu clade, specifically in various prophages, e.g., PSA-like, A118-like and A500-like.

**Figure 2 microorganisms-09-01323-f002:**
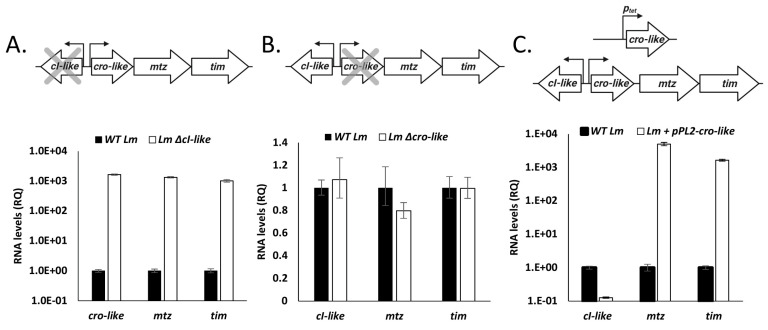
The *cI-cro*-like locus possesses the transcriptional dynamics of a phage regulatory switch. RT-qPCR transcriptional analysis of the *cI-cro*-like locus transcripts obtained from WT *Lm* grown to mid-log phase in BHI at 37 °C, and from bacteria deleted for the *cI*-like gene (**A**), deleted for the *cro*-like gene (**B**), or ectopically expressing the *cro*-like gene under regulation of p*_tetR_* promoter (**C**). Data are presented as relative quantity (RQ) compared to the mRNA levels in WT bacteria, normalized to the levels of the *rpoD* gene. The data represent three independent experiments. Error bars indicate the 95% confidence interval.

**Figure 3 microorganisms-09-01323-f003:**
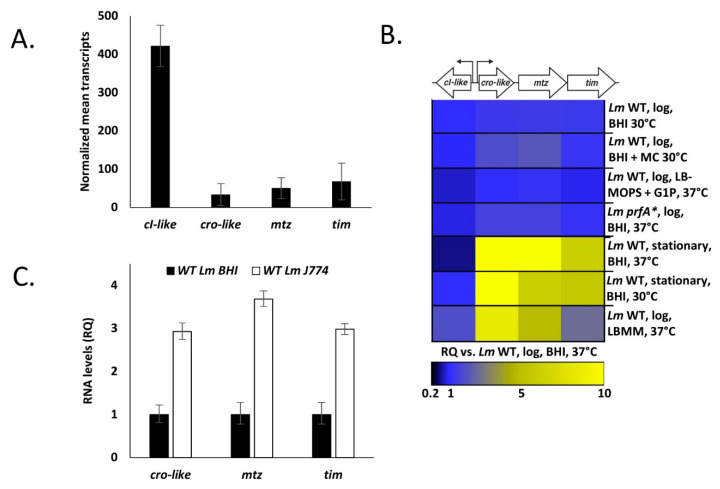
A *cro-like, mtz* and *tim* expression is induced during the stationary phase of growth and during growth in intracellular or intracellular-like conditions. (**A**) Quantitative analysis of the transcription level of the *cl-cro-like* locus of WT *Lm* grown to mid-log phase in BHI at 37 °C. Transcriptional levels are normalized to the transcription level of the *rpoD* gene. The data represent the mean of three independent experiments. Error bars indicate standard deviation. (**B**) Transcriptional analysis of the *cIcro-like* locus in WT Lm grown in different conditions, and of *prfA** mutant strain. RT-qPCR data are presented as relative quantity (RQ), compared to the mRNA levels in WT bacteria grown to mid-log phase in BHI at 37 °C and normalized to the levels of the *rpoD* gene. The data are presented as a heat map and incorporate three independent experiments. (**C**) RT-qPCR transcriptional analysis of the *cI-cro-like* locus of WT *Lm* grown in J774 cells for 6 h. Data are presented as relative quantity (RQ), compared to the levels in WT bacteria and normalized to the levels of the *rpoD* gene. The data represent two independent experiments. Error bars indicate the 95% confidence interval.

**Figure 4 microorganisms-09-01323-f004:**
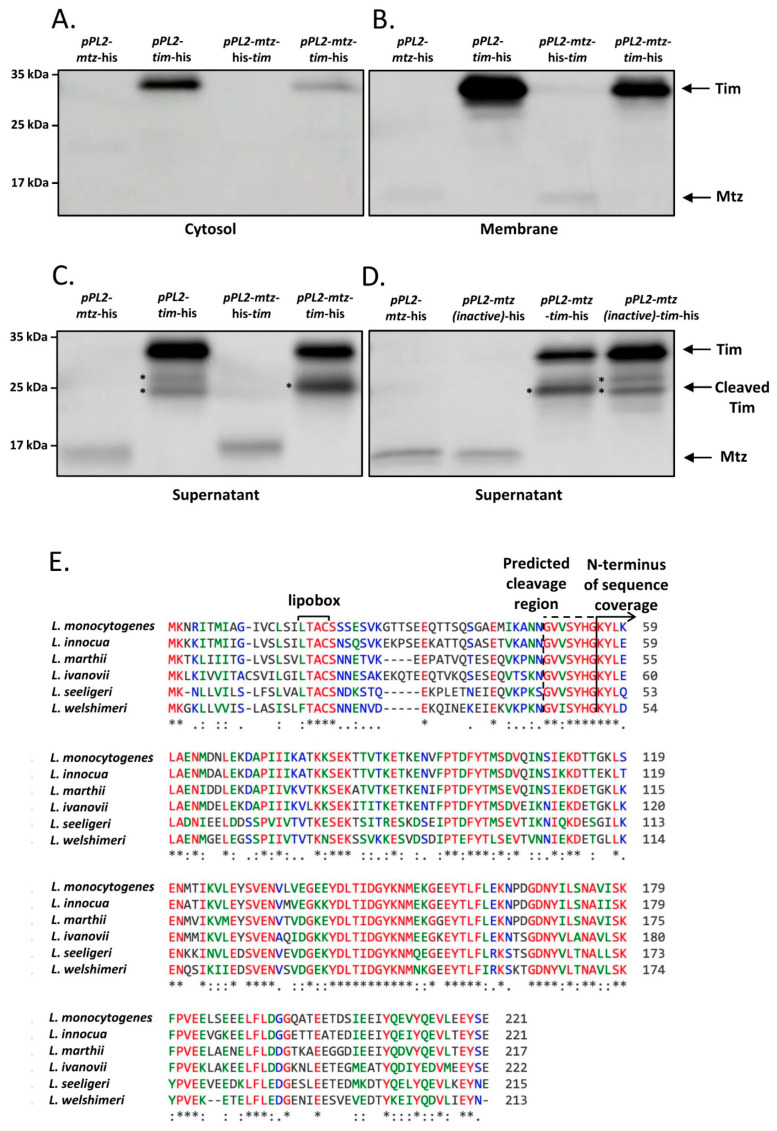
Tim is cleaved by Mtz. Western blot analysis of Tim and Mtz proteins 6His tagged at C-terminus, obtained from indicated bacterial strains grown to mid-log phase in BHI at 37 °C. Total proteins were separated into the cytosolic (**A**), membrane (**B**) and supernatant fractions (**C**,**D**). Equal amounts of proteins in each fraction were separated on 12.5% SDS-PAGE, blotted and probed with anti-6His antibody. The experiment was performed 3 times, and the figure shows a representative blot. Tim cleavage products are marked with asterisks. (**E**) Multiple alignments of the Tim homologues of the clade *Listeria* sensu strictu species. The conserved N-terminal lipobox is indicated. The N-terminus of the sequence coverage of the purified cleaved Tim-His is marked by a plain line, and the predicted cleavage site is marked by a broken line. The following accession numbers were used for the alignment: AEO06250.1 (corresponds to gene tag LMRG_00714, *L. monocytogenes* strain 10403S); CAC96535.1 (*L. innocua*); EFR87930.1 (*L. marthii*); CBW85704.1 (*L. ivanovii*); MBC1942631.1 (*L. seeligeri*); CAK20663.1 (*L. welshimeri*). Deduced amino acid sequences were aligned using the ClustalW program (https://npsa-prabi.ibcp.fr/cgi-bin/npsa_automat.pl?page=/NPSA/npsa_clustalw.html, accessed on 1 June 2021).

**Figure 5 microorganisms-09-01323-f005:**
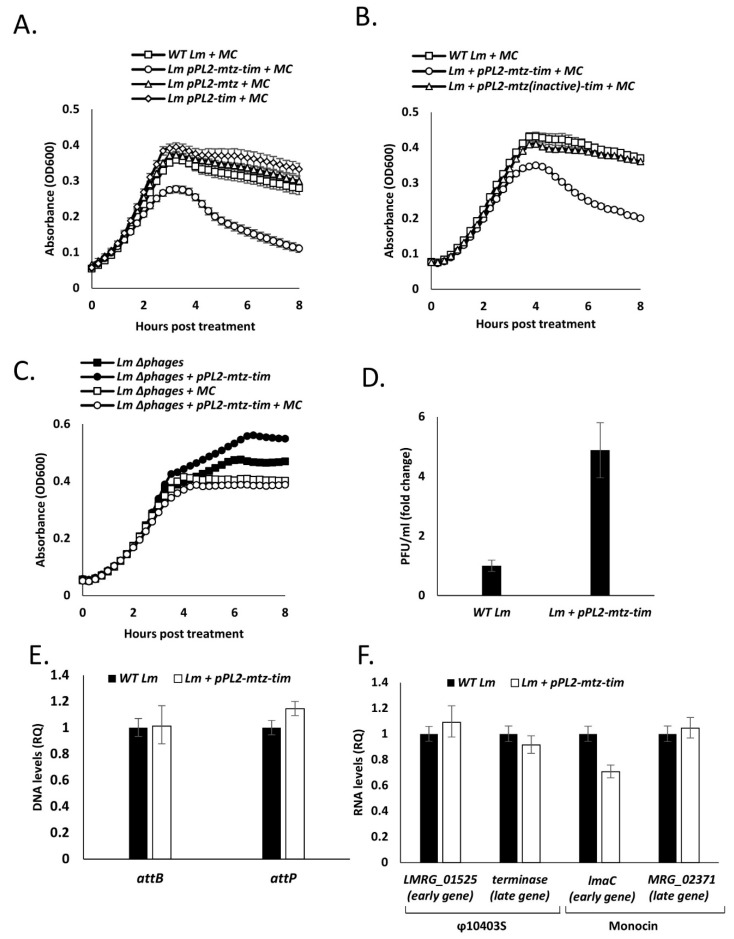
Ectopic expression of Mtz-Tim enhances phage-mediated lysis. (**A**) Growth analysis in BHI media of WT *Lm*, and WT *Lm* harboring pPL2 integrative plasmid expressing *mtz, tim* or both from inducible p*_tetR_* promoter (pPL2-*mtz,* pPL2-*tim*, and pPL2-*mtz*-*tim* respectively) with addition of MC at 37 °C. (**B**) Growth analysis in BHI media of WT *Lm*, and WT *Lm* harboring pPL2 integrative plasmid expressing *mtz*-*tim* or *mtz* mutated at position E136A with *tim* from an inducible p*_tetR_* promoter (pPL2-*mtz*-*tim* and pPL2-*mtz(inactive)*-*tim*, respectively) with addition of MC at 37 °C. (**C**) Growth analysis in BHI media of WT *Lm*, and deletion mutant lacking prophage elements (Δ*phages*) harboring pPL2 plasmid expressing *mtz-tim* from an inducible p*_tetR_* promoter (pPL2-*mtz-tim*) with and without addition of MC, as indicated, at 37 °C. Data in A-C represent three independent experiments and error bars represent the standard deviation of triplicates. (**D**) Production of infective virions assayed as PFU in WT *Lm* bacteria and WT *Lm* harboring pPL2 integrative plasmid expressing *mtz-tim* from an inducible p*_tetR_* promoter (pPL2-*mtz-tim*) grown at 37 °C in BHI with MC for 5 h. The results are normalized to PFU of WT bacteria. The data are the mean of three independent experiments and error bars represent the standard deviation. (**E**) Phage excision and replication levels in WT *Lm* and WT *Lm* harboring pPL2 integrative plasmid expressing *mtz-tim* from inducible p*_tetR_* promoter (pPL2-*mtz-tim*). RT-qPCR analysis of intact *comK* gene, representing the φ 10403S *attB* site, and the circularized phage genome, representing the *attP* site, performed on DNA extracted from bacteria grown at 37 °C in BHI with MC for 4 h. Data presented as relative quantity (RQ), compared to the levels in WT bacteria and normalized to the levels of the *rpoD* gene. The data represent three independent experiments and error bars indicate a 95% confidence interval. (**F**) RT-qPCR transcriptional analysis of the indicated φ10403S and monocin genes of *Lm* bacteria deleted for phage lysis factors and the same strain harboring pPL2 integrative plasmid expressing *mtz-tim* from inducible p*_tetR_* promoter (pPL2-*mtz-tim*) grown at 37 °C in BHI with MC for 4h. Data are presented as relative quantity (RQ), compared to the levels in WT bacteria and normalized to the levels of the *rpoD* gene. The data represent three independent experiments. Error bars indicate the 95% confidence interval.

**Figure 6 microorganisms-09-01323-f006:**
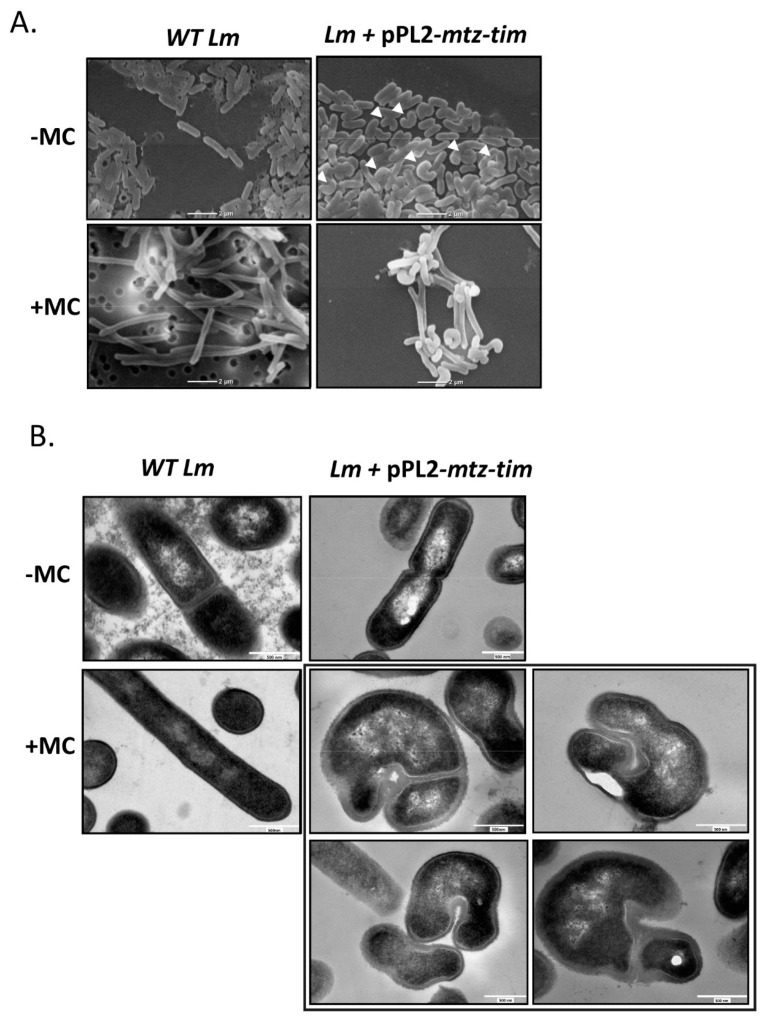
Ectopic expression of Mtz-Tim affects *Lm* morphology. (**A**) Scanning electron microscopy images of WT *Lm* bacteria and WT *Lm* harboring pPL2 plasmid expressing *mtz-tim* from an inducible p*_tetR_* promoter (pPL2-*mtz-tim*) grown at 37 °C in BHI with or without MC for 4 h. Experiment was performed 3 times and a representative image is shown. Bacteria with curved morphology indicated with white arrowheads. (**B**) Transmission electron microscopy images of WT *Lm* bacteria and WT *Lm* harboring pPL2 plasmid expressing *mtz-tim* from an inducible p*_tetR_* promoter (pPL2-*mtz-tim*) grown at 37 °C in BHI with or without MC for 4 h. The gray box contains four images of *Lm* expressing *mtz-tim* treated with MC. Experiment was performed 2 times and a representative image is shown.

**Figure 7 microorganisms-09-01323-f007:**
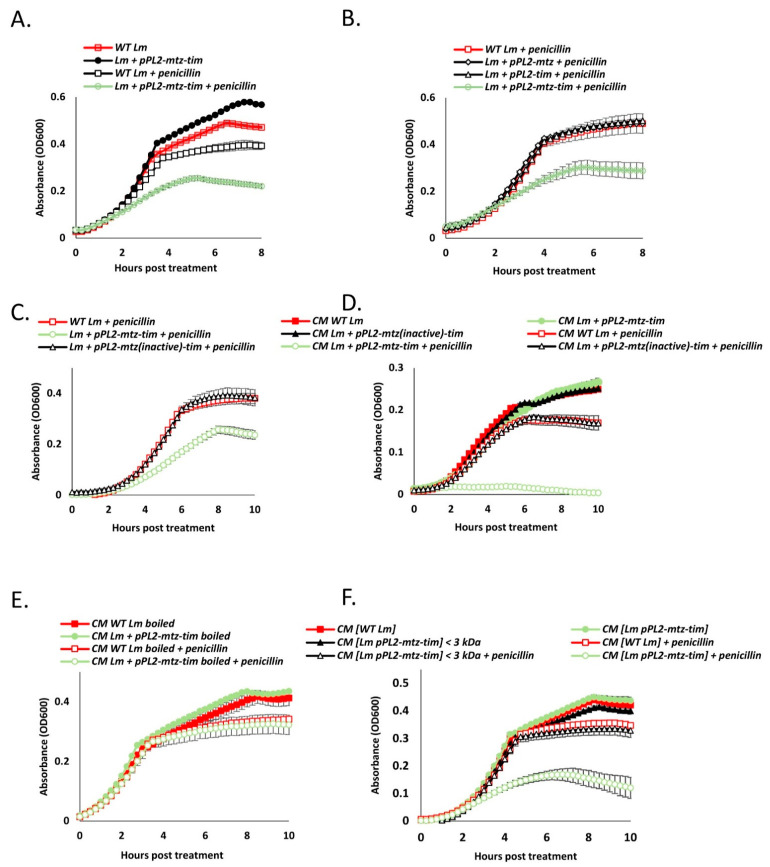
Ectopic expression of Mtz-Tim sensitize *Lm* to penicillin. (**A**) Growth analysis in BHI media of WT *Lm*, and WT *Lm* harboring pPL2 plasmid expressing *mtz-tim* from an inducible p*_tetR_* promoter (pPL2-*mtz-tim*) at 37 °C, with and without addition of penicillin, as indicated. (**B**) Growth analysis in BHI media of WT *Lm*, and WT *Lm* harboring pPL2 plasmid expressing *mtz* or *tim* or both from an inducible p*_tetR_* promoter (pPL2-*mtz*, pPL2-*tim* and pPL2-*mtz-tim*, respectively) at 37 °C, with the addition of penicillin. (**C**) Growth analysis in BHI media of WT *Lm*, and WT *Lm* harboring pPL2 plasmid expressing *mtz-tim* or *mtz* mutated at position E136A with *tim* from an inducible p*_tetR_* promoter (pPL2-*mtz-tim* and pPL2-*mtz(inactive)-tim*, respectively) at 37 °C, with the addition of penicillin. (**D**) Growth analysis of WT *Lm* in conditioned media (CM) of WT *Lm*, and WT *Lm* expressing *mtz-tim* or *mtz* mutated at position E136A with *tim* from an inducible p*_tetR_* promoter (pPL2-*mtz-tim* and pPL2-*mtz(inactive)-tim*, respectively) at 37 °C, with or without addition of penicillin. (**E**) Growth analysis of WT *Lm* in conditioned media (CM) or in boiled CM of WT *Lm* or WT *Lm* expressing *mtz*-*tim* from an inducible p*_tetR_* promoter (pPL2-*mtz-tim*), with or without addition of penicillin at 37 °C. (**F**) Growth analysis of WT *Lm* in conditioned media (CM) or in CM depleted of proteins >3 kDa (CM < 3kD) of WT *Lm* or WT *Lm* expressing *mtz*-*tim* from an inducible p*_tetR_* promoter (and pPL2-*mtz-tim*) at 37 °C, with or without addition of penicillin. Data in A–F represent three independent experiments and error bars represent the standard deviation of triplicates.

**Figure 8 microorganisms-09-01323-f008:**
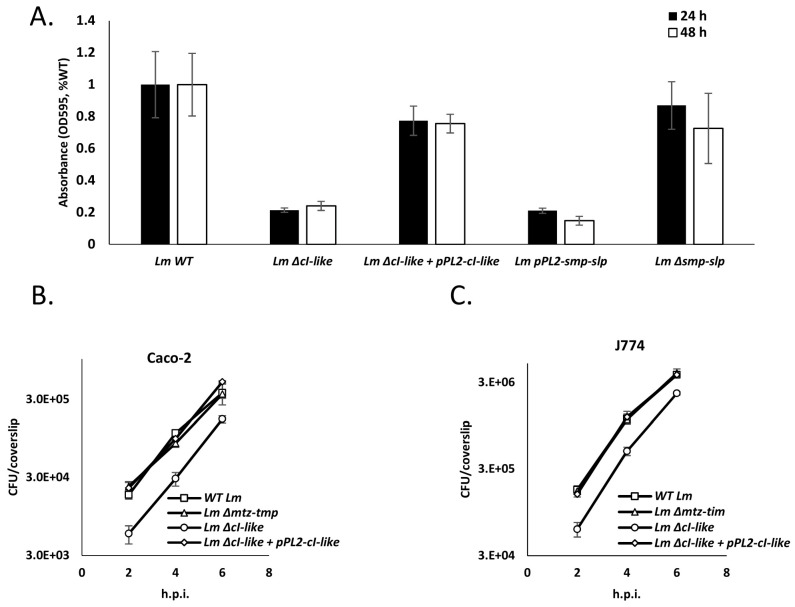
Ectopic expression of Mtz-Tim hinders mammalian cells invasion and biofilm production. (**A**) Biofilm formation analysis of WT *Lm*, WT *Lm* harboring pPL2 integrative plasmid expressing *mtz-tim* from an inducible p*_tetR_* promoter (pPL2-*mtz-tim*), Δ*mtz-tim*, Δ*cI*-like and Δ*cI*-like harboring pPL2 plasmid expressing *cI-*like from an inducible p*_tetR_* promoter (pPL2-*cI-*like) following 24 h of incubation in BHI at 37 °C without shaking. The data represent three independent experiments, error bars represent the standard deviation of triplicates. (**B**) Intracellular growth analysis of WT *Lm*, and Δ*mtz-tim*, Δ*cI-*like and Δ*cI-*like harboring pPL2 plasmid expressing *cI-*like from an inducible p*_tetR_* promoter (pPL2-*cI-*like) in J774 or Caco-2 (**C**) cells. The same number of bacteria were used to infect the cells. The data represent three independent experiments, error bars represent the standard deviation of triplicates.

## Data Availability

The authors declare that all data supporting the findings of this study are available within the paper and its Appendix A or from the authors upon reasonable request.

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
