# Peer review of "A Metzincin and TIMP-Like Protein Pair of a Phage Origin Sensitize Listeria monocytogenes to Phage Lysins and Other Cell Wall Targeting Agents"

_microorganisms, 2021, doi:10.3390/microorganisms9061323_

Round 1

Reviewer 1 Report

In the manuscript “A Metzincin and TIMP-like protein pair of a phage origin sen-sitize Listeria monocytogenes to phage lysins and other cell wall targeting agents” of Etai Boichis and colleagues, the authors reported on a phage remnant DNA region that might play a role in the lifecycle of the bacteria. The authors found a locus consisting of two XRE regulators and a genes coding for a secreted metzincin protease and a TIMP-family like metzincin inhibitor. The XRE regulators were found to act as prophage repressor/Cro repressor proteins regulating the expression of the metzincin and TIMP-like genes. The products seemed to alter the bacterial cell morphology and increase its sensitivity to phage mediated lysis. Furthermore, the proteins also seemed to have an effect on the modulation of the cell wall structure.

The provided study was carefully conducted and the manuscript is well written. The data presented her provide novel insights into the Lm lifestyle. The manuscript can be published without further improvements. That is a really great study – congratulation to the authors – well done!!!

I would be interested in the regulation of the cI and cro gene – Did you also find the conserved  lambda operator sequences for the regulation or are they new?

Author Response

I would be interested in the regulation of the cI and cro gene – Did you also find the conserved  lambda operator sequences for the regulation or are they new?

No, we don’t find classic operator sites as appear in lambda CI-Cro regulatory region. There are two sites that share some similarity, but since we did not examine them, we don’t feel confident to present them. That said, this should be explored.

We thank this reviewer for his/her interest.

Reviewer 2 Report

The study conducted by Boichis et al. sought to evaluate the role of Metzincin and TIMP-like proteins of viral origin in Listeria. The manuscript is well written, and the experimental procedures clearly designed and with a purpose. In the end, it is appreciable the amount of information provided and the new proposed mechanisms. In general, my questions refer to the following points:

  1. Why the effect of the ectopic expression of mtz-tim on Lm morphology was also not evaluated in the mutant cured of its two phage elements (Δphages)? I observed this information only for the penicillin sensitivity assay.
  2. Why was the biofilm test performed only in 24 hours? Since several experiments considered the stationary phase in this study, I would expect longer incubation times for the biofilm assay.
  3. Why did the authors evaluate the effect of penicillin only on the Lm growth curve? Do the authors believe that greater sensitivity to penicillin could also reflect in lower MIC values for this antibiotic? What would be the implication of this finding for other classes of antibiotics?
  4. In the discussion section the authors mention that the pair Mtz-Tim "... cause Lm (both the bacteria expressing them and their neighbors in the culture medium) to be more sensitive to a variety of lytic agents: phage holin-lysin, penicillin, mutanolysin and Triton X-100 ". At the community level, do the authors believe that a higher concentration of free Tim could impact by sensitizing other taxa?

Author Response

1. Why the effect of the ectopic expression of mtz-tim on Lm morphology was also not evaluated in the mutant cured of its two phage elements (Δphages)? I observed this information only for the penicillin sensitivity assay.

Indeed, we missed this experiment. Upon the reviewer request, we wished to do it now, however we learned that there is a long waiting list for the TEM, and it will take 2-3 months to get the data. That said, looking at figure 6A, one can see that some of the Mtz-Tim expressing bacteria are curved, even without MC treatment, a condition in which the two phage elements are fully repressed. Based on this observation we expect a similar result in the Δphages mutant.

2. Why was the biofilm test performed only in 24 hours? Since several experiments considered the stationary phase in this study, I would expect longer incubation times for the biofilm assay.

Following the reviewer comment, we performed an additional biofilm formation assay at 48 h, and the results were similar to that presented at 24 h. The new data was added to Figure 8A.

3. Why did the authors evaluate the effect of penicillin only on the Lm growth curve? Do the authors believe that greater sensitivity to penicillin could also reflect in lower MIC values for this antibiotic? What would be the implication of this finding for other classes of antibiotics?

Following the reviewer comment, we preformed additional experiments to evaluate the MIC values of penicillin in Lm bacteria and in bacteria overexpressing mtz-tim. We were not able to detect a change in the MIC value in the concentration range of 0.0015-0.5 ug/ml. As for the second question, we evaluated the effect of mtz-tim overexpression on the sensitivity to two additional cell-wall targeting antibiotics; bacitracin and D-cycloserine, and found similar results as with penicillin. The new data is shown in Figure S4.

4. In the discussion section the authors mention that the pair Mtz-Tim "... cause Lm (both the bacteria expressing them and their neighbors in the culture medium) to be more sensitive to a variety of lytic agents: phage holin-lysin, penicillin, mutanolysin and Triton X-100 ". At the community level, do the authors believe that a higher concentration of free Tim could impact by sensitizing other taxa?

This is a very intriguing possibility that Mtz-Tim impact other taxa. We really don’t know. It will be nice to examine that in the future.

We thank this reviewer for his/her interest and important comments.

Reviewer 3 Report

Excellent paper for microorganisms, worthy of publishing as is.

Just some minors before publication:

keywords formatting; need high resolution figures to replace fig 1 and 4; in figure 7, use diiferent colors to label different group curves would be clearer.

Author Response

keywords formatting- We formatted the keywords.

need high resolution figures to replace fig 1 and 4; in figure 7- High resolution figures are now included.

Use different colors to label different group curves would be clearer- we used different colors in Fig.7.

We thank this reviewer for his/her interest and comments.